# Core Signs and Symptoms in Children with Autism Spectrum Disorder Improved after Starting Risperidone and Aripiprazole in Combination with Standard Supportive Therapies: A Large, Single-Center, Retrospective Case Series

**DOI:** 10.3390/brainsci12050618

**Published:** 2022-05-09

**Authors:** Hamza A. Alsayouf, Haitham Talo, Marisa L. Biddappa

**Affiliations:** Kids Neuro Clinic and Rehab Center, Dubai Healthcare City, Al Razi Medical Complex, Dubai 1015, United Arab Emirates; htalo@yahoo.com (H.T.); marisa.lobobiddappa@gmail.com (M.L.B.)

**Keywords:** autism spectrum disorder, risperidone, aripiprazole, antipsychotic agents, case study

## Abstract

**Background:** There are a number of medications prescribed to address comorbid challenging behaviors in children with autism spectrum disorder (ASD), including risperidone and aripiprazole. This retrospective case series reports the use of these drugs in children aged 2 to 13 years. **Methodology:** A total of 82 children (mean age, 5 years; 79% male) with ASD treated at the Kids Neuro Clinic and Rehab Center in Dubai between January 2020 and September 2021 were included in this retrospective case series. All patients had comorbid challenging behaviors that were resistant to standard supportive therapies alone and warranted pharmacological intervention. The Childhood Autism Rating Scale—2nd Edition Standard form (CARS2-ST) and the Clinical Global Impression (CGI)—Severity (CGI-S) and CGI—Improvement (CGI-I) scales were used to assess the severity of ASD at baseline and to monitor response to treatment with risperidone or aripiprazole. **Results:** Besides the expected improvement in comorbid challenging behaviors, 79/82 patients (96%) attained a CGI-I score of 2 or 1 following treatment, and 35/82 patients (43%) achieved both a CGI-I score of 1 and minimal-to-no symptoms as per the CARS2-ST test, with complete resolution of their ASD signs and symptoms. The differences in the overall mean CARS2-ST and CGI-S scores pre- and post-treatment were statistically significant (Z = −7.86, *p* < 0.0001 for both), with pre- and post-treatment mean values of 42 and 23 for CARS2-ST, respectively, and 6 and 2 for CGI-S, respectively. The main side effects were asymptomatic elevated prolactin (*n* = 12) and excessive weight gain (*n* = 2). **Conclusions:** ASD core symptoms and comorbid behaviors in young children improved following chronic treatment with antipsychotic medications, either with or without medications for attention deficit hyperactivity disorder, when combined with standard supportive therapies. Double-blind, placebo-controlled clinical trials are needed to verify these findings.

## 1. Introduction

Autism spectrum disorder (ASD) is a multifaceted neurodevelopmental condition with a unique presentation in each patient [1,2]. Besides the presence of restricted, repetitive, and stereotyped behaviors, ASD is characterized by difficulties with social interactions and interpersonal relationships. In 2013, the Diagnostic and Statistical Manual of Mental Disorders, 5th Edition (DSM-5) combined four types of autism, which were previously seen as separate, under the umbrella term of ‘autism spectrum disorder’ [1]. Autistic disorder, Asperger’s syndrome, childhood disintegrative disorder, and pervasive developmental disability not otherwise specified are now all classified as a form of ASD [1,2].

Children with ASD exhibit a wide range of symptoms that can appear before the age of 2 years but are most evident between 2 and 3 years of age [1,3]. Several symptoms can persist in later adulthood despite improvement in the core domains of ASD; these include restricted interests, reduced social smiling, and limited emotional expressiveness [4,5,6,7]. Notably, a 40-year longitudinal study of patients with ASD revealed overall improvement in young adulthood but a decline at a later age [4]. Providing treatment early in life may offer children with ASD the best chance to achieve healthy development and long-term improvement [2,8].

Early intervention with developmental, occupational, and speech therapies to promote healthy development and improve social skills is the standard treatment strategy for ASD in children [2]. No regulatory agency has yet approved a drug to treat the core signs and symptoms of ASD [9]. Instead, to improve social functioning and foster development, medications are used as an adjunctive therapy to address comorbid challenging behaviors [10,11]. The FDA authorized the use of risperidone in 2006 and aripiprazole in 2009 for children aged 5 and 6 years and older, respectively, to assist with the comorbid behaviors associated with ASD [12,13]. Due to the possible adverse effects of atypical antipsychotic medications, which include drowsiness, weight gain, and metabolic abnormalities, the use of these medications must be closely monitored [9,14]. Further, medications used to treat attention deficit hyperactivity disorder (ADHD), such as methylphenidate, atomoxetine, and bupropion, have been found to reduce hyperactivity and enhance awareness in children with ASD [15,16,17,18,19,20,21]; these medications are also known to assist in the management of excessive weight gain by suppressing the appetite.

Here, we present a retrospective case series of 82 children aged 2–13 years with ASD and the associated comorbid behavioral difficulties that were refractory to standard supportive therapies. These children were treated with risperidone or aripiprazole, either with or without atomoxetine or methylphenidate to boost attention and maintain weight, in conjunction with standard supportive therapies. There was significant clinical improvement in the children’s comorbid behaviors as well as their ASD core signs and symptoms based on the Childhood Autism Rating Scale—2nd Edition Standard form (CARS2-ST) and the Clinical Global Impression (CGI)—Severity (CGI-S) and CGI—Improvement (CGI-I) scales [22,23]. 

## 2. Case Presentation

### 2.1. Patients and Data Collection

This retrospective case series comprised patients treated at the Kids Neuro Clinic and Rehab Center in Dubai between January 2020 and September 2021, and who met the following criteria: (1) diagnosed with ASD as per the DSM-5 criteria and the CARS2-ST (and not the CARS2-ST High Functioning) test; (2) followed the treatment regimen for ≥6 months within this time period; (3) ≥2 years and <18 years of age at the time of starting the treatment regimen; (4) comorbid behavioral issues that were resistant to standard supportive therapies and warranted pharmacological intervention, such as aggression, insomnia, anxiety, ADHD, depression, or self-mutilation; (5) normal electroencephalogram with sleep sample; (6) normal auditory brainstem response (ABR) hearing test; (7) no other associated chronic medical condition; (8) no findings suggestive of a genetic disorder (such as dysmorphic features, a strong family history of genetic disorders, or an abnormal genetic test); (9) no associated global developmental delay; (10) CGI-S score ≥4 at the time of presentation; (11) no previous treatment with any antipsychotic medication; (12) the family was compliant with the treatment protocol with regular visits every 1 to 2 months; (13) the patient’s parents/guardian provided informed consent; and (14) the patient had follow-up at the time of data collection (the data collection endpoint for the purpose of this study was 30 September 2021).

The following data were collected for all patients based on chart review: age at presentation, age at treatment initiation, medications and doses used, adverse effects, laboratory analyses performed, initial formal diagnosis per the DSM-5 criteria and CARS2-ST, their baseline CARS2-ST and CGI-S scores before starting medications, and their CGI-I score and final CARS2-ST and CGI-S scores at their last visit to our clinic. 

### 2.2. Ethics Approval

This study was performed in accordance with the Declaration of Helsinki (1964) and approved by the Dubai Healthcare City Regulatory Ethics Committee: Approval no. KNCRC-02.

### 2.3. Diagnosis and Monitoring

When a patient with signs and symptoms suggestive of ASD presented at our clinic, we performed a thorough history and physical examination. In addition, we performed an electroencephalogram with a sleep sample to evaluate for electrical status epilepticus in sleep or epileptic encephalopathy, as well as an ABR test to assess for hearing loss. These tests were conducted to exclude potentially treatable conditions requiring different treatment approaches [24,25,26,27]. Pharmacological treatment was recommended if the individual had a normal ABR, normal electroencephalogram, and met the above-mentioned criteria. In each case, the diagnosis was obtained by both a pediatric neurologist and a clinical psychologist following a multidisciplinary assessment and diagnostic approach.

The CGI-S and CARS2-ST were used as a baseline to rate each patient’s severity prior to initiating medication. Severity was scored on the seven-point CGI-S scale: 1 = normal, not at all ill; 2 = borderline, mentally ill; 3 = mildly ill; 4 = moderately ill; 5 = markedly ill; 6 = severely ill; 7 = among the most extremely ill patients [22]. CARS2-ST can be applied to children aged 2 to 6 years, as well as to older children with an estimated IQ ≤79 or a notable communication impairment; it a 15-category scoring system to classify patients into one of three ASD severity groups, namely minimal-to-no symptoms of ASD, mild-to-moderate symptoms of ASD, and severe symptoms of ASD [23,28,29,30].

The CGI-I scale was used to monitor for potential improvement in each patient after treatment initiation. CGI-I is scored based on a seven-point scale that compares the patient’s current and baseline condition (initiation of treatment): 1 = very much improved since the initiation of treatment; 2 = much improved; 3 = minimally improved; 4 = no change from baseline; 5 = minimally worse; 6 = much worse; 7 = very much worse since the initiation of treatment [22]. If a patient attained a CGI-I score of 1 or 2, a clinical evaluation and repeat CGI-S and CARS2-ST tests were performed to confirm this improvement.

### 2.4. Treatment

We used the same treatment protocol as that published previously [31,32]. Patients were prescribed either risperidone or aripiprazole [12,13]. Because risperidone has been more extensively studied among children with ASD, it was suggested as the first-line treatment; however, several families opted for aripiprazole instead. The children’s parents and legal guardians were counseled regarding the off-label use and potential side effects of these antipsychotic medications. All medications were given as daily doses. This study was a retrospective case series; therefore, no control (placebo) treatment or randomization was performed. 

Risperidone was typically initiated with 0.25 mg orally at night and gradually increased every 1 to 2 weeks to the maximum dose of 3 mg or the maximum dose and schedule tolerated. This approach was initiated for 28/82 cases. Aripiprazole treatment patients were started on 0.5 mg at night and this dose was gradually increased every 1 to 2 weeks to the maximum dose (15 mg at night) or the maximum dose tolerated. The regimens for risperidone and aripiprazole were adhered to while the patient showed improvement with each dose increment and experienced no adverse effects. 

When a patient tolerated their medication without adverse effects and with progressive improvement to a CGI-I score of 1 or 2, the medication was continued. If their CGI-I plateaued at a score of 2 or higher, or there was excessive weight gain and persistent hyperactivity or inadequate attention span, methylphenidate or atomoxetine was added to augment improvement to a CGI-I score of 1, improve attention, and control excessive weight gain. Further, all children with progressive weight gain were referred to a nutritionist to minimize the development of other health issues and premature weaning off of their medication. Methylphenidate was usually added at 2.5 mg in the morning and increased gradually to a maximum total daily dose of 20 mg or less as tolerated. Atomoxetine was typically started at 10 mg at night and increased gradually to a maximum dose of 2 mg per kg or less as tolerated if no adverse effects were observed and an improvement was seen with each dose increment. Clonidine was prescribed if the patient continued to show hyperactivity or aggression despite using the above-mentioned medications [33,34,35,36]. Escitalopram was tried if the patient continued or started to show anxiety or obsessive-compulsive behavior despite using the above-mentioned medications [37,38]. If a patient reached a CGI-I score of 1 and a repeat CARS2-ST score of no symptoms, the medications were continued for a further 6 months before attempting to slowly wean off all medications. 

All patients were strongly encouraged to continue with standard supportive therapies such as speech therapy, applied behavior analysis (ABA) therapy, and occupational therapy. Parents and caregivers were given practical support and guidance and were advised to limit screen time and increase their child’s opportunities for social interaction by enrollment in a daycare facility, nursery, or playgroup.

### 2.5. Monitoring for Safety

Before therapy initiation, a complete blood count, kidney function test, and lipid panel were performed, and the levels of electrolytes, liver enzymes, hemoglobin A1c, and prolactin were measured to generate a baseline against which any metabolic side effects of the antipsychotic medications could be monitored [39,40,41,42,43]. Tests were repeated after 3 to 6 months, with subsequent yearly checkups. A fully trained pediatric cardiologist performed an echocardiogram and electrocardiogram on all patients when the maximum dose was reached or earlier if indicated [44,45,46]. At every office visit, serial measurements of weight, body mass index (BMI), height, and vital signs were obtained to further monitor for any side effects. Further, the Abnormal Involuntary Movement Scale (AIMS) was completed at every visit for each patient [47].

### 2.6. Statistical Analysis

Differences between the ordinal data in this study (pre- and post-treatment CGI-S and CARS2-ST scores) were evaluated with the Wilcoxon signed-rank test. A two-tailed *p*-value < 0.05 was considered statistically significant. Data were analyzed using the VassarStats online suite of statistical tools (www.vassarstats.net), accessed on 1 December 2021.

## 3. Outcomes

### 3.1. Patient Disposition

There were 142 children with ASD treated with aripiprazole or risperidone at our center between January 2020 and September 2021, but only 82 children met the criteria for inclusion in this retrospective case series: 25 patients were lost to follow up, 26 patients were noncompliant as they did not have regular follow-up, and 9 patients had an associated global developmental delay or genetic disorder. Of the 82 children with ASD included in this case study, 79% were male, and all were aged between 2 and 13 years (mean age, 5 years) at presentation; 28 families (34% of patients) decided to initiate treatment with risperidone, whereas 54 families (66% of patients) chose aripiprazole (Table 1). 

### 3.2. Treatment Outcomes

Besides the expected improvement in comorbid challenging behaviors, the treatment strategy employed here resulted in 79/82 patients (96%) attaining a CGI-I score of 2 or 1, and 35/82 patients (43%) achieved both a CGI-I score of 1 and minimal-to-no symptoms as per the CARS2-ST test (i.e., a score of <30), along with complete resolution of their ASD signs and symptoms as per clinical evaluation. One of these patients is currently off all medication, while the remainder are still under treatment. The differences in the overall mean CARS2-ST and CGI-S scores pre- and post-treatment were statistically significant (Z = −7.86, *p* < 0.0001 for both) using the Wilcoxon signed-rank test, with pre- and post-treatment mean values of 42 and 23 for CARS2-ST, respectively, and 6 and 2 for CGI-S, respectively. The pre- and post-treatment scores for the two separate treatment groups are shown in Table 2.

In the aripiprazole group, 45/54 patients (83%) were diagnosed with severe autism and 9/54 patients (17%) with mild-to-moderate autism based on the CARS2-ST scale. Of the 45 patients with severe ASD at the start of medication, 36/45 (80%) showed substantial improvement to minimal-to-no-symptoms. Of the remaining patients with severe ASD, 8/45 (18%) improved from severe to mild-to-moderate, whereas 1/45 (2%) stayed in the severe range although there was an improvement in his overall score (from 47 to 39). All 9 of the aripiprazole-treated patients diagnosed with mild-to-moderate autism at the beginning of treatment improved to minimal-or-no symptoms. In the aripiprazole group, 24/54 patients (43%) were treated with aripiprazole only, whereas 9/54 required atomoxetine, 16/54 used methylphenidate with aripiprazole, 3/54 used methylphenidate and atomoxetine in addition to aripiprazole, 1 patient required escitalopram with atomoxetine and aripiprazole for persistent anxiety and obsessive-compulsive behavior, and 1 patient required clonidine and aripiprazole for persistent hyperactivity. Medication dose ranges in this group were as follows: aripiprazole 1.5–15 mg, methylphenidate 2.5–20 mg, atomoxetine 10–60 mg, escitalopram 10 mg, and clonidine 0.1 mg. 

In the risperidone group, 23/28 patients (82%) were diagnosed as having severe autism and 5/28 (18%) as having mild-to-moderate autism based on the CARS2-ST scale. Of the patients with severe ASD, 21/23 (91%) showed substantial improvement to minimal-to-no symptoms. The remaining 2/23 (9%) severe cases in the risperidone group improved to mild-to-moderate severity. All 5 of the patients in the risperidone group diagnosed with mild-to-moderate autism improved to minimal-to-no symptoms. Eight patients (6/28) required only risperidone, 7/28 used risperidone and methylphenidate, 7/28 used risperidone and atomoxetine, 6/28 needed risperidone, atomoxetine, and methylphenidate, 1 required clonidine and risperidone for persistent aggression, and 1 was prescribed risperidone and escitalopram for persistent obsessive-compulsive behavior. In the risperidone group, 27/28 patients are still on medication and 1 patient has already weaned off medication with a CARS2-ST score of 15. Medication dose ranges in this group were as follows: risperidone 1–3 mg, methylphenidate 2.5–20 mg, atomoxetine 10–60 mg, escitalopram 10 mg, and clonidine 0.1 mg.

### 3.3. Safety Outcomes

The main side effects noted for risperidone and aripiprazole were asymptomatic elevated prolactin and weight gain. Twelve patients (aripiprazole group, *n* = 2; risperidone group, *n* = 10) with a median age of 5 years (range, 2–6) developed asymptomatic elevated prolactin levels that were treated conservatively following consultation with a pediatric endocrinologist. Weight gain (increase in BMI to >25) was excessive in 2 patients (*n* = 1 in each treatment group); this was managed by consultations with a nutritionist, dietary changes, and the addition of methylphenidate or atomoxetine where required to improve attention and hyperactivity, as these medications also suppress the appetite. All other laboratory findings were within normal limits for the patients’ age range, and cardiac evaluations did not uncover any adverse cardiac effects. Serial AIMS tests were performed at each visit to monitor for any medication-induced abnormal movements, but no patient developed any such abnormalities. In some patients, methylphenidate caused irritability and atomoxetine caused aggression, and these medications were then discontinued. Other side effects included mild sedation and drooling that were mainly observed at the start of treatment and later resolved with the continuation of treatment.

## 4. Discussion

The current study supports the findings of our two previously published case series [31,32], in which the chronic use of risperidone or aripiprazole, along with standard supportive therapies and with or without ADHD medication (methylphenidate or atomoxetine), was associated with improvement in the core signs and symptoms of ASD in young children. The CGI-I score improved in 79/82 patients (96%) to 2 or 1, with 35/82 patients (43%) experiencing complete remission of their ASD symptoms (CGI-I score of 1 and CARS2-ST score of <30) after receiving early and chronic therapy of at least 6 months in duration. This is the first large retrospective case series (*n* = 82) to indicate that ASD core signs and symptoms can be treated successfully. Our two earlier case series documented the adoption of a comparable treatment strategy for 18 and 10 children aged 2 to 10 years old and demonstrated the complete resolution of ASD symptoms in 56% and 60% of patients, respectively. Although several previous studies had reported some improvement in ASD core signs and symptoms in young children following treatment, none have reported complete remission (these studies are reviewed in our previous case series) [48,49,50,51,52,53]. Notably, while ASD signs and symptoms can in some cases improve without any intervention, studies indicate that a clear diagnosis in younger children tends to remain stable over the short term (1 to 2 years) [54], suggesting that the improvements observed here were not incidental. 

Here, risperidone and aripiprazole were prescribed to children with ASD at a younger age (from 2 years) than what is currently approved by the FDA (5 and 6 years, respectively) to assist with challenging behaviors that were proving a major barrier to learning, were resistant to standard supportive therapies alone, and warranted pharmacological intervention. This treatment regimen was offered to patients whose parents or legal guardians consented to the regimen, and who understood the potential adverse effects and the off-label use of these medications. Thus, we combined non-biologic (e.g., ABA therapy) and biological therapies to treat children with autism in a region where the vast majority of families cannot afford regular treatment with the recommended range of supportive therapies (ABA therapy, speech therapy, and occupational therapy). 

In addition to previous studies showing improvement in ASD core signs and symptoms in young children following treatment with risperidone or aripiprazole [48,49,50,51,52,53], a number of studies have shown that the use of these medications can aid learning and cognitive development in children with ASD [55,56,57,58,59,60,61]. Unfortunately, however, the majority of the studies performed on ASD in children thus far have been short-term, employed prescription drugs to chiefly regulate behavioral problems, have not combined antipsychotic medications with ADHD medications, and have not included a cross-over arm for non-responders [52,55,56,57,58,59,60,62,63,64,65,66,67].

Because attention is critical in the learning process, we augmented concentration with ADHD medications as required. The co-occurrence of ADHD in young children with ASD further impacts on their function and quality of life [28], and the beneficial effects of ADHD medications on anxiety, inattention, social communication, and self-regulation in this population is supported by a number of previous studies and reports [15,16,17,18,19,20,21]. 

In our patients, the side effects following medication were either slight and fleeting (such as drowsiness and drooling), asymptomatic (elevated prolactin), or treatable with adjunctive treatment such as dietary changes or the use of methylphenidate or atomoxetine for both inattention and excessive weight gain. However, methylphenidate and atomoxetine were omitted wherever possible since these medications may increase the risk of adverse effects and their consumption in very young children is unapproved. Both risperidone and aripiprazole have previously been linked to obesity and asymptomatic elevated prolactin [9,40,61,68,69], and these were the main side effects observed in our patients. 

There are several limitations inherent in this retrospective case series: it was open-label without controls or randomization; the confounding effects of non-pharmacological interventions were not, therefore, controlled for; and the patients did not follow a unified treatment model. The age and IQ for some children in this study exceeded the ranges for which the CARS2-ST was designed, which might have resulted in an underestimation of autistic symptoms for those children. Despite the fact that no definitive conclusions can be drawn owing to the study’s limitations, we expect that our findings will inspire further research in this area.

## 5. Conclusions

Following chronic treatment with aripiprazole or risperidone, the majority of the 82 children aged 2–13 treated in this retrospective case series showed a marked improvement in their core ASD signs and symptoms to a score of 1 or 2 on the CGI-I scale following an individualized treatment approach, with almost half experiencing complete remission. These findings suggest that antipsychotic medications used early in childhood may be able to eliminate the core signs and symptoms of ASD in a significant proportion of patients when combined with ADHD medications as needed following and individualized treatment approach. To validate these outcomes in the pediatric population, properly powered double-blind and placebo-controlled trials need to be conducted.

## Figures and Tables

**Table 1 brainsci-12-00618-t001:** Patient demographics and baseline disease characteristics.

	Aripiprazole Group (*n* = 54)	Risperidone Group (*n* = 28)	Total (*n* = 82)
**Age, years**			
Mean ± SD	5.4 ± 2.4	4.3 ± 1.8	5.0 ± 2.3
Range	2–13	2–10	2–13
**Male, *n* (%)**	41 (76)	24 (86)	65 (79)
**ASD severity at baseline, *n* (%)**			
Mild-to-moderate	9 (17)	5 (18)	14 (17)
Severe	45 (83)	23 (82)	68 (83)

ASD, autism spectrum disorder; SD, standard deviation.

**Table 2 brainsci-12-00618-t002:** ASD core signs and symptoms at baseline and following treatment with aripiprazole or risperidone.

	Baseline	After Treatment
**Aripiprazole (*n* = 54)**		
CARS2-ST score	42 ± 6	24 ± 6
CGI-S score	6 ± 1	3 ± 1
CGI-I score	–	2 ± 1
Severe ASD symptoms, *n* (%)	45 (83)	1 (2)
Minimal-to-no ASD symptoms, *n* (%)	–	45 (83)
**Risperidone (*n* = 28)**		
CARS2-ST score	41 ± 5	23 ± 5
CGI-S score	6 ± 1	2 ± 1
CGI-I score	–	2 ± 1
Severe ASD symptoms, *n* (%)	23 (82)	0
Minimal-to-no ASD symptoms, *n* (%)	–	26 (93)

Values represent the mean ± standard deviation unless otherwise indicated. ASD, autism spectrum disorder.

## Data Availability

Not applicable.

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
