# Peer review of "Core Signs and Symptoms in Children with Autism Spectrum Disorder Improved after Starting Risperidone and Aripiprazole in Combination with Standard Supportive Therapies: A Large, Single-Center, Retrospective Case Series"

_brainsci, 2022, doi:10.3390/brainsci12050618_

Round 1
Reviewer 1 Report
The manuscript entitled, “Risperidone and Aripiprazole in Children with Autism Spectrum Disorder Substantially Improves Core Signs and Symptoms in Combination with Standard Supportive Therapies: A 4 Large, Single-Center, Retrospective Case Series” describes the efficacy of the two drugs in managing ASD children even under the age of 6 years although below six years are not yet FDA approved. The study evaluated 82 children with ASD and report that the drugs were well tolerated with a significant outcome.
This is a retrospective review of the cases treated between January 2020 to September 2021, although throughout the manuscript, the narration does not reflect that it was a chart review and a retrospective study. The reader may consider this study as a preliminary report that intentionally tested the two drugs in underage (below 6 years) children to observe how well the drugs were tolerated if so, please clarify clearly.
Reading through the manuscript, it is clear that this is a well-planned preliminary study to report the efficacy of the two drugs, Risperidone and Aripiprazole. While there have been few clinical trials being conducted in children under the age of 6, the description of the study appears to be a preliminary study conducted to evaluate its efficacy. It clearly reveals that all precautions were taken to administer the drugs, and the study report is based on the recent treatment strategy, reasonably administered as a preliminary trial. Hence, I would suggest the authors rephrase the title to preliminary report rather than retrospective case series.
The title can read, “Risperidone and Aripiprazole in Children with Autism Spectrum Disorder Substantially Improves Core Signs and Symptoms in Combination with Standard Supportive Therapies: A 4 Large, Single-Center, preliminary study”
Pg2 ln57-58: In the sentence, “The FDA authorized the use of risperidone in 2006 and 57 aripiprazole in 2009 for children aged 5 and 6 years and older, respectively,” – please change to “aged 6 years and older”, since the FDA approval is for only for children 6 years and older.
Pg2 ln 59-60: In the sentence, “Due to the possible adverse effects 59 of these atypical antipsychotics,” change to atypical antipsychotic medications.
Pg3 ln129-131: In the sentence, “Because risperidone has been 129 more extensively studied among children with ASD, it was suggested as the first-line 130 treatment; however, several families opted for aripiprazole instead.” Please provide the reason why the families were opting for aripiprazole. Was it intentionally suggested to the family? Or, do they realize risperidone had more side effects if they were administered for a prolonged period.
Please provide a detailed legend for table 2. It is not clear how the groups were analyzed to report the numbers. Are the numbers corresponding to the baseline for after treatment? It would be helpful for the readers if the authors can explain the outcome of the after-treatment.
It would be helpful for the readers to group the children aged 2 to 6 as one group and above 6 as another group to evaluate the dose tolerance for the two drugs.
Author Response
I have attached the revised manuscript for reviewer to see changes we made based on their recommendations.
Comment 1:
The manuscript entitled, “Risperidone and Aripiprazole in Children with Autism Spectrum Disorder Substantially Improves Core Signs and Symptoms in Combination with Standard Supportive Therapies: A 4 Large, Single-Center, Retrospective Case Series” describes the efficacy of the two drugs in managing ASD children even under the age of 6 years although below six years are not yet FDA approved. The study evaluated 82 children with ASD and report that the drugs were well tolerated with a significant outcome.
This is a retrospective review of the cases treated between January 2020 to September 2021, although throughout the manuscript, the narration does not reflect that it was a chart review and a retrospective study. The reader may consider this study as a preliminary report that intentionally tested the two drugs in underage (below 6 years) children to observe how well the drugs were tolerated if so, please clarify clearly.
Response:
We thank the reviewer for their feedback. We agree that our study does have elements of a prospective study in that we prospectively planned to later collate and analyze the data that is now reported in this manuscript. However, data were indeed collected retrospectively by chart review and there were no controls included. As with many retrospective case series, it is hoped that our findings will later lead to large, prospective, randomized, and controlled trials. For clarification, we have now added a statement to the Methods that data collection in this retrospective case series was based on chart review (Line 93).
Comment 2:
Reading through the manuscript, it is clear that this is a well-planned preliminary study to report the efficacy of the two drugs, Risperidone and Aripiprazole. While there have been few clinical trials being conducted in children under the age of 6, the description of the study appears to be a preliminary study conducted to evaluate its efficacy. It clearly reveals that all precautions were taken to administer the drugs, and the study report is based on the recent treatment strategy, reasonably administered as a preliminary trial. Hence, I would suggest the authors rephrase the title to preliminary report rather than retrospective case series.
The title can read, “Risperidone and Aripiprazole in Children with Autism Spectrum Disorder Substantially Improves Core Signs and Symptoms in Combination with Standard Supportive Therapies: A 4 Large, Single-Center, preliminary study”.
Response:
Please see our response to Comment 1 above. We believe that the current title is the most appropriate given the nature of the study. However, we are open to the Editors’ guidance on this matter.
Comment 3:
Pg2 ln57-58: In the sentence, “The FDA authorized the use of risperidone in 2006 and 57 aripiprazole in 2009 for children aged 5 and 6 years and older, respectively,” – please change to “aged 6 years and older”, since the FDA approval is for only for children 6 years and older.
Response:
Please note that our original statement here is correct (risperidone was approved for use in children aged 5 years and older), based on the reference cited in our manuscript (FDA approval letter for risperidone: https://www.accessdata.fda.gov/drugsatfda_docs/nda/2006/020272Orig1s036,s041,020588Orig1s024,s028,s029,21444Orig1s008,s015.pdf) and other information availably widely online regarding risperidone (https://medlineplus.gov/druginfo/meds/a694015.html).
Comment 4:
Pg2 ln 59-60: In the sentence, “Due to the possible adverse effects 59 of these atypical antipsychotics,” change to atypical antipsychotic medications.
Response:
We have made this change (Line 59).
Comment 5:
Pg3 ln129-131: In the sentence, “Because risperidone has been 129 more extensively studied among children with ASD, it was suggested as the first-line 130 treatment; however, several families opted for aripiprazole instead.” Please provide the reason why the families were opting for aripiprazole. Was it intentionally suggested to the family? Or, do they realize risperidone had more side effects if they were administered for a prolonged period.
Response:
In contrast with aripiprazole, risperidone is widely used in the Middle East. Many families read the parents’ reviews of the two medications online before deciding which to initiate treatment with. The more widespread use of risperidone in this region has led to more widely reported side effects and this appears to be why most families move away from this medication. However, we did not systematically record the reasons underlying the families’ choice, and we, therefore, cannot provide a definitive reason for this trend.
Comment 6:
Please provide a detailed legend for table 2. It is not clear how the groups were analyzed to report the numbers. Are the numbers corresponding to the baseline for after treatment? It would be helpful for the readers if the authors can explain the outcome of the after-treatment.
Response:
We thank the reviewer for this helpful comment. We have rewritten the heading for Table 2 and amended some of the row descriptors for increased clarity.
Comment 7:
It would be helpful for the readers to group the children aged 2 to 6 as one group and above 6 as another group to evaluate the dose tolerance for the two drugs.
Response:
Please note that Table 2 only describes treatment effectiveness and does not provide any data on dose tolerance. To address this comment, we have now provided the median and age range of the patients in whom the main side effect was noted, namely asymptomatically elevated prolactin (Results, Line 247).

Reviewer 2 Report
Interesting article, however these drugs are usually prescribed to treat ASD co-symptoms. Here, the authors demonstrated ASD minimal to no symptoms in several patients, after treatments. Did these patients not more autistic ones? This point is very important
As Major issues:
- even if the authors wrote a brief statement in the limitations, more should be due on the possibility that also other co- treatments could have an effect. How is possible to exclude them?
- -add the treatment was daily.
Author Response
I have attached revised manuscript so the reviewer can see changes we made based in his recommendations.
Comment 1:
Interesting article, however these drugs are usually prescribed to treat ASD co-symptoms. Here, the authors demonstrated ASD minimal to no symptoms in several patients, after treatments. Did these patients not more autistic ones? This point is very important
Response:
Patients who achieved a CGI-I score of 1 and a CARS2-ST of <30 displayed minimal-to-no symptoms during clinical evaluation and are no longer considered autistic. One of these patients is off all medication now.
Comment 2:
Even if the authors wrote a brief statement in the limitations, more should be due on the possibility that also other co- treatments could have an effect. How is possible to exclude them?
Response:
Please note that we have already described this as a limitation in the Discussion: “There are several limitations inherent in this retrospective case series: it was open-label without controls or randomization; the confounding effects of non-pharmacological interventions were not, therefore, controlled for; and the patients did not follow a unified treatment model.” (Lines 309–312).
Comment 3:
Add the treatment was daily
Response:
We thank the reviewer for this comment. We have now included this information to the Treatment section of the Methods (Line 134).

Round 2
Reviewer 2 Report
Authors well answered to my comments.
Author Response
Thanks for reviewers.
